# Genesis of Significance of Carbonated Thermal Water Springs in Xining Basin, China

Yude Lei [1,2,3] , Zhen Zhao [1,2,3], Baojian Zhang [4,5,*], Xianchun Tang [4,5], Yinfei Luo [1,2,3], Guiling Wang [4,5,*], Jun Gao [4,5,*] and Dailei Zhang [4,5]

[1] Key Laboratory of Geo-Environment Qinghai Province, Xining 810007, China
[2] Environmental Geological Prospecting Bureau of Qinghai Province, Xining 810007, China
[3] Qinghai 906 Engineering Survey and Design Institute Co., Ltd., Xining 810007, China
[4] Chinese Academy of Geological Sciences, Deep Earth Science and Exploration Technology Laboratory, Beijing 100037, China
[5] Technology Innovation Center of Geothermal and Hot Dry Rock Exploration and Development, Ministry of Natural Resources, Shijiazhuang 050061, China
* Correspondence: zbjsddk@126.com (B.Z.); guilingw@163.com (G.W.); gjun@cags.ac.cn (J.G.)

**Abstract:** There are 30 carbonate hot springs in Yaoshuitan geothermal field, Xining Basin, China, with a temperature of 18~41.5 °C; and there are 10 carbonate hot springs in Qijiachuan geothermal field, with a temperature of 10~19.5 °C. Both geothermal fields are carbonate hot springs containing large amounts of $CO_2$ gas. In order to reveal the origin of the carbonated hot springs in Yaoshuitan and Qijiachuan of Xining Basin, this paper offers a comprehensive study of the regional deep geology, tectonic setting, total analysis of carbonated hot springs, $\delta^2H$, $\delta^{18}O$, $\delta^{13}C$ isotopes, main gas composition, and geochemical characteristics of travertine dating, travertine $\delta^{13}C$, and rare earth elements. The geological process of carbonated hot spring formation and the evolution of $H^+$ content from deep to shallow is revealed, and the genetic mechanism of the carbonated hot spring in Xining Basin is systematically summarized. The results show that: (1) The characteristics of $\delta^2H$ and $\delta^{18}O$ isotopes indicate that the recharge source of carbonated thermal water springs in Xining Basin is mainly atmospheric precipitation. The age of carbonated thermal water springs at $^{14}C$ is more than 20 ka, indicating that some of them may come from deep fluid (gas) sources. The R/Ra in carbonated thermal water springs is mostly less than 1, indicating that the helium in geothermal water is mainly crustal source helium, and there is no deep mantle source material. (2) The Piper three-plot indicates that the direction of groundwater evolution from the recharge area at the edge of Xining Basin to Yaoshuitan and Qijiachuan carbonated thermal water spring area near the edge of the basin is opposite to the normal path of groundwater evolution in the basin, which is due to the large amount of $CO_2$ gas mixed in the deep fault along the northern margin of Laji Mountain. The ratio of $(Ca^{2+} + Mg^{2+})$ and $(HCO_3^- + SO_4^{2-})$ in the Potan and Qijiachuan carbonated thermal water springs is close to 1, and the ratio of $(Na^+ + K^+)/HCO_3^-$ is less than 1. It indicates that the chemical composition of the Yaoshuitan carbonated thermal water spring and the Qijiachuan carbonated thermal water spring in Xining Basin is dominated by the dissolution of calcite, dolomite, and gypsum in deep carbonate reservoirs, supplemented by the dissolution of silicate minerals. The relationship between the volume fraction of $CO_2$ and the $\delta^{13}C$ value of carbon isotope of $CO_2$ indicates that the source of $CO_2$ is inorganic, which is mainly formed by metamorphism and decomposition of deep carbonate and marble. The $\delta Eu < 1$ and $\delta Ce > 1$ of the rare earth elements in the calcium center of the carbonated thermal water springs indicate that the groundwater supplying the travertine material has been in the acidic environment receiving $CO_2$ from the deep crust for a long time. (3) A series of tectonic activities, such as late collision and post-collision between the Indian and Eurasian plates, has led to the uplift, asthenosphere upwelling, and thermal invasion of the northern Tibetan Plateau and other deep dynamic processes. The deep faults in the northern margin of the Laji Mountain and other deep faults with obvious neotectonic activity have provided channels for the up-invasion of deep thermal materials, and local geothermal anomalies were formed near the deep faults. The hidden carbonate rocks and silicate rocks with large thickness undergo thermal metamorphism under high temperature

and high pressure in the deep geothermal anomaly area and form a large amount of $CO_2$, which is dissolved in water and enhances the acidity of water. At the same time, the dissolution reaction of acidic water to carbonate rocks consumes $H^+$, which keeps the carbonated thermal water spring weakly acidic. (4) The composition of travertine in carbonated thermal water springs is dominated by calcite, indicating that travertine may be formed in a deep geological environment with a temperature of 150~200 °C, indicating that there are abnormal heat sources in shallow carbonate strata with a burial depth of 3000~4000 m. The abnormal heat source may be caused by the deep fault in the northern margin of Laji Mountain, as well as other deep and large faults channeled in the deep crust and mantle heat source, indicating that the deep fault in the northern margin of Laji Mountain has an obvious heat-controlling effect, and there is a good prospect of geothermal resources exploration near the fault.

**Keywords:** carbonated thermal water springs; genetic mechanism; deep fault in the northern margin of Laji Mountain; heat control; Xining Basin

## 1. Introduction

The Earth's deep-source gas emissions [1] and their environmental effects [2,3] have been gradually attracting the attention of geologists. Most of the carbon in the Earth system occurs in the Earth's interior [4], and is mainly released from the Earth's interior to the atmosphere in the form of $CO_2$ through different tectonic processes [5–7]. In recent years, studies on deep-source gas emissions, including $CO_2$ from volcanic–geothermal systems, as well as deep faults and their causes, have increased [8]. The $CO_2$ degassing flux and genesis of the Lassa terrane–Yadong rift volcanic–geothermal system in southern Tibet have been studied [9,10].

Springs with carbonated geothermal water have a special type of geothermal water. In both neotectonic active areas and modern volcanic active areas, a large amount of deep-source $CO_2$ is mainly released outward through geothermal water as a carrier, and only a small amount is released via geothermal springs in the form of $CO_2$ gas. This is because $CO_2$ is easily dissolved in water; in particular, under conditions of deep, high confining pressure, $CO_2$ usually migrates and is released with geothermal water [11]. It can be seen that carbonated geothermal springs provide an important outlet for $CO_2$ emissions in the Earth. Current studies on carbonated geothermal springs mainly focus on the relationship between earthquakes and the distribution of carbonated thermal water springs [11,12]. Research on carbonated thermal water springs and their formation mechanisms has increased in recent years, which indicates that the study of fluid earth is garnering increasing attention. Chris et al. (2018) studied the Valles Caldera geothermal system in the Nacimiento Mountains and pointed out that deep circulation along faults and mixing of different aquifers are responsible for the formation of carbonated waters springs [13]. Nisi et al. (2019) studied the $CO_2$-rich fluid geochemistry in the Campo de Calatrava volcanic area in Spain, and pointed out that the interaction between shallow water and deep carbon dioxide caused a decrease in pH [14]. Italiano et al. (2016) studied the geochemical characteristics and genesis of dissolved gasses in the eastern Carpathian Mountains, Transylvania Basin, Romania, and pointed out that in addition to obvious atmospheric gas sources, volcanic and crustal gas sources provide materials for circulating groundwater [15]. Shvartsev et al. (2017) analyzed the origin and evolution of the high $pCO_2$ groundwater in the Mukhen geothermal springs in the Russian Far East [16]. Kharaka et al. (2018), through an experiment on the reaction between groundwater and $CO_2$, showed that the leakage of $CO_2$ into groundwater increased acidity, which increased the content of Fe and other metals in groundwater [17]. Zhang et al. (2005) and Li et al. (2018) studied the genesis of carbonated thermal water spring $CO_2$ in the Hengjing area of southern Jiangxi Province and the Heyuan fault zone of Guangdong Province, respectively, and pointed out that the carbonated thermal water spring $CO_2$ in Hengjing area of southern Jiangxi Province is

mainly deep mantle-derived inorganic gas related to deep and large fault activities [18]. The carbonated thermal water spring $CO_2$ in the Heyuan fault zone is of mixed origin, from mantle source and metamorphism, mainly the latter [19]. Zhou et al. (2020) studied the geochemical characteristics of hot spring gas in the Jinshajiang–Honghe fault zone and pointed out that $CO_2$ gas in geothermal springs mainly came from limestone cracking [20]. Yu et al. (2022) studied the geochemical characteristics and geological significance of geothermal spring gas in the northern Yadong Rift Valley of the Qinghai Tibet Plateau, and pointed out that the thermal decomposition of carbonate rocks was the main source of $CO_2$ [21]. Walter et al. (2020) studied $CO_2$ released into the atmosphere by geothermal springs in the Sperchios basin, northern Greece, and the resulting travertine deposition [22]. Wang et al. (2020) pointed out that due to the high $pCO_2$ values (10–3.5 atm) in geothermal spring water and the escape of $CO_2$ from water after the spring water emerged from the surface, $CaCO_3$ was deposited to form travertine [23]. Ta et al. (2020) discussed the evolution and source of major ions in triassic carbonate hot springs in Chongqing. These related studies revealed the source and migration of $CO_2$ gas in hot springs from different perspectives [24].

Xining Basin is located in the northeast of Qinghai Province. The study of Xining geothermals began in the 1960s. In recent years, due to the increasing degree of geothermal development and utilization, Xining geothermal research has gradually attracted scholarly attention. There has been significant research data collected on the geothermal geology of Xining Basin that mainly focus on geothermal field characteristics [25] and causes [26,27], water chemistry characteristics and causes [28,29], a thermal storage conceptual model [30], etc. The Yaoshuitan geothermal spring in Xining Basin has been developed and utilized for hundreds of years, and is called a 'sacred spring' because of its obvious therapeutic effects. In 1986, the Qinghai New Energy Research Institute carried out an evaluation of drinking mineral water of the Yaoshuitan spring group, and the free $CO_2$ content of two springs in Yaoshuitan spring group was found to be as high as 718.5~884.4 mg/L. At the end of 2002, the China Geological Survey discovered a high $CO_2$ gas–water mixed high-pressure artesian thermal well in the terrace of Qijiachuan in Pingan County, Qinghai Province. In addition, in 2016, when China Railway First Survey and Design Institute Group Co., Ltd. constructed the exploration hole of the Xining–Chengdu Railway Haidongnanshan Tunnel (about 5 km from the Qijiachuan), gas at a depth of ~5 m and a strong whistling sound was drilled out. The gas had no abnormal smell. It took 11 days to complete the drilling and plugging, and there was no attenuation. After borehole sampling and analysis, the gas composition was mainly $CO_2$ [31]. It was proved that there are carbonated thermal water springs and $CO_2$ gas reservoirs in parts of Xining Basin. Zheng et al. (2016) studied the accumulation model of the $CO_2$ gas reservoir in the Pingan area of Qinghai Province [32]. These exploration results prove that there are carbonated thermal water springs and $CO_2$ gas reservoirs in some parts of Xining Basin, and they are weakly acidic. Although the formation of $CO_2$ gas reservoirs in Xining Basin has been preliminarily revealed, the formation of carbonated thermal water springs and the reason why carbonated thermal water springs are weakly acidic have not been systematically studied.

In this study, detailed hydrochemistry and isotope studies were conducted on carbonated thermal water springs, ordinary geothermal water, and gas samples collected from the Xining Basin. The purpose of this study is: (i) to determine the origin and hydrogeochemical formation process of carbonated thermal water springs, and to preliminarily analyze the heat source; (ii) to analyze the origin of $CO_2$ in carbonated thermal water springs and the changes of $H^+$ content during its migration from the deep to the surface; (iii) to reveal the relationship between the formation of carbonated thermal water springs and the deep fault in the northern margin of Laji Mountain, judge the heat control function of the deep fault in the northern margin of Laji Mountain, and deepen the understanding of the occurrence law of geothermal resources in this area.

## 2. Geological Background

### 2.1. Regional Geological Features

Xining Basin is located in the Middle Qilian island arc uplift zone, which is a Meso-Cenozoic faulted basin. The formation of Xining Basin is restricted by the deep faults of the north margin of Laji Mountain in the south and the deep faults of Daban Mountain in the north (Figure 1). Xining Basin comprises cap rocks and a basement. The cap rocks are Mesozoic and Cenozoic stratigraphic strata, which are mainly characterized by the large thickness of sedimentary rocks, basically horizontal in occurrence and with a large distribution range. The basement structure mainly includes three-level structural units, such as the Laoyeshan uplift, western slope, Shuangshuwan depression, Dabaozi–Xining uplift, Zongzhai depression, and Xiaoxia uplift. Xining Basin has an obvious double-layer structure, and its basement is a dome structure composed of Proterozoic. Above the basement are the Triassic, continental Jurassic, fluvial–lacustrine Cretaceous, inland lacustrine Paleogene, piedmont alluvial–proluvial facies Neogene and Quaternary, and contact with false conformity or angle unconformity. The basement morphology of Xining Basin is not only uneven, but generally shows the characteristics of surrounding elevation and descending to the middle. In the south–north tectonic profile, Xining Basin also has the geothermal tectonic background of deep thermal uplift, extrusion above the neutral tectonic plane, the southern and northern orogenic belts thrust toward the basin, and peristaltic expansion under the 'pull-down and up-pressure'.

In the north of Xining Basin, there are the Daban Mountains with an altitude of 4000~6000 m, and in the south, there is Laji Mountain with an altitude of more than 4000 m. Under the action of regional principal compressive stress of NEE, geological units on both sides of Xining Basin thrust toward the middle, resulting in fold deformation and fault activity of the basin. The basin edge fault zones that affect the deformation of the basin mainly include the Laji Mountain fault zone, the Dabanshan fault zone, and the Riyueshan fault zone. These faults have obvious activities in the Quaternary period. The Laji Mountain fault zone and the Dabanshan fault zone are characterized by sinistral compression movements, and the Riyueshan fault is mainly characterized by dextral movements. The near north–south faults in the basin are characterized by tension–tensional torsion, high angle, strong activity during the neotectonic period, seismogenesis, water conduction, and heat control.

### 2.2. Geothermal Geological Characteristics

Xining Basin is a closed confined artesian water basin, and the Mesozoic and Cenozoic clastic sediment particles change from coarse to finer from the edge to center of the basin. In the mountainous area at the edge of the basin, after the groundwater is recharged by atmospheric precipitation and surface water, part of the groundwater runs off along the structural fissures to the center and deep part of the basin, and the deep circulating groundwater is heated in the deep part of the basin to form geothermal water. Because of the continuous circulation movement of groundwater, the deep heat energy is brought to the shallow part, which is discharged in the form of geothermal springs and underground runoff to the outside of the basin. Therefore, Xining Basin is both an artesian water basin and an geothermal water artesian basin [33]. The Meso–Cenozoic fissure-pore-type geothermal reservoir in the interior of the basin and the carbonate fissure-karst-cavity geothermal reservoir in the orogenic belt around the basin are the large-scale and mining-significant thermal storage in Xining Basin. Since the Cenozoic, influenced by the Himalayan orogeny, Xining Basin has experienced strong neotectonic movements, manifested in the resurrection of old faults around the basin and within the basin, and further intensified deep hydrothermal activities, which created a regional geological structure background for the formation of geothermal resources in Xining Basin.

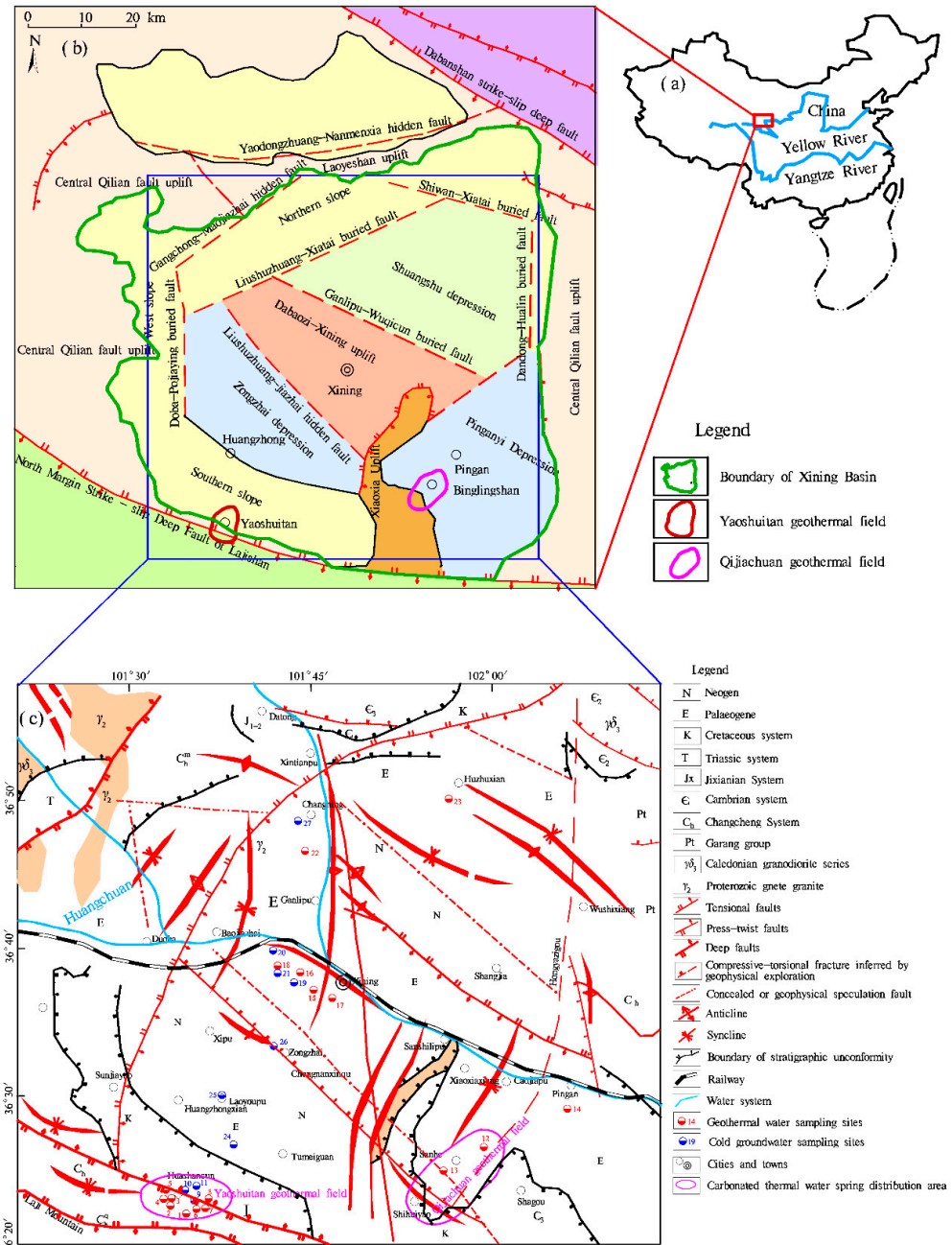

**Figure 1.** Sketch of geological structure and sampling points and fault location in Xining Basin (slightly modified from reference [27]).

The Yaoshuitan geothermal field is located in the southern margin of Xining Basin. The geomorphology is the northern slope of Laji Mountain and the middle and upper reaches of the Nanchuan Valley. The tectonic location belongs to the northern edge of the Laji Mountain fault zone of the Qilian Mountains' fold system. The northern margin of Laji Mountain is a deep and large crustal fault zone with a good convective geothermal structure, and its fracture zone is thicker; according to the drilling, there is a fractured zone with a thickness of 1177 m between depths of 103 and 1280 m. Ground mineral water is linearly arranged in the form of springs along the NE direction of the Shuitan fault, forming a large area of marshes and wetlands. The outcrops of the hot field are mainly: Proterozoic gray-white sericite mellite, light gray, black-gray mellite sand–slate; Quaternary alluvial–proluvial sand and gravel; and Quaternary chemical deposits, mainly sinter. Quaternary alluvial–proluvial sand–gravel thickness is 2.0–20.6 m, and its pores are mostly filled with travertine, so it is

cemented sand–gravel. Quaternary chemical deposits (sinter) spread throughout the mineral water distribution area, with a general thickness of 0.8 to 1.5 m, forming a plate-like covering on the ground. The basic aquifers of geothermal wells and geothermal springs are carbonate rocks developed by huge thick fissure karst under the cap rocks. There are 30 natural springs in the mineral water area. The temperature of the springs is 18~41.5 °C, and the free $CO_2$ content is 6.12~884 mg/L. The geothermal spring water spews out of the surface in the form of a mixture of water and vapor. The pH of geothermal spring water is 6.12~6.55, and it is weakly acidic. The hydrochemical types are mainly $HCO_3$-Ca·Mg and $HCO_3$-Ca, and the salinity is 993–1850 mg/L. The cold spring water is slightly alkaline, the pH is 7.10~7.69, and the salinity is generally less than 500 m deep.

The Qijiachuan geothermal field is located in the Ping'an Depression to the east of Xiaoxia uplift in Xining Basin, and stratigraphic distribution is controlled by the deep fault zone in the north margin of Laji Mountain and Xining Basin. The Upper Proterozoic, Paleozoic, Mesozoic, and Cenozoic strata are all outcropped in the area. The Upper Proterozoic outcropping area is small and sporadic, forming a regional crystalline basement. The Mesozoic and Cenozoic in the area are mainly distributed in the Mesozoic Cretaceous sandstone and conglomerate, and the Cenozoic Paleogene and Neogene mudstone and siltstone. Quaternary aeolian, alluvial, ice-water deposits, and a small amount of travertine deposits are widely distributed, most of which have weak consolidation, loose structures, and varying degrees of thickness. There are 10 geothermal mineral springs, mainly distributed in the NW and NW-trending active fault belts in the piedmont of Laji Mountain, controlled by the geological structure, and there are generally travertine deposits near the spring vent. The geothermal spring water is ejected to the surface as a mixture of water and air, and the partial pressure of $CO_2$ in the water sample at the spring vent or well head is 59,238.6~71,340 Pa. The pH of geothermal spring water is 6.54~6.63, showing weak acidity. The chemical types of water are mainly $HCO_3$·$SO_4$-Ca·Na and $HCO_3$-Ca·Mg·Na, and the salinity is 2940~5470 mg/L. The higher salinity of geothermal water in this area is due to a larger amount of gypsum salt in the Cretaceous system.

## 3. Sample Analysis and Research Methods

### 3.1. Sample Collection and Analysis

In order to determine the origin of carbonated thermal water springs in Xining Basin, according to the occurrence characteristics of geothermal water in Xining Basin, the carbonated thermal water springs in Yaoshuitan geothermal field, Huangzhong County, Qijiachuan geothermal field, Pingan County, and other geothermal water with normal $CO_2$ content in Xining Basin were analyzed. A total of 29 samples of geothermal water and groundwater were collected; 14 geothermal water samples were analyzed, including 8 Yaoshuitan geothermal spring samples, 2 Qijiachuan geothermal spring samples, and 4 geothermal water from other parts of Xining Basin for comparison. A total of 10 samples of groundwater analysis were analyzed, including 3 samples of Yaoshuitan groundwater, Xining Basin, and 7 samples from other parts of Xining Basin. Isotopic analysis of $\delta^2H$ and $\delta^{18}O$ was performed on 5 geothermal water samples, including 2 samples of Yaoshuitan geothermal spring, 1 samples of Qijiachuan geothermal spring, and 2 samples of geothermal water from other parts of Xining Basin for comparison (Figure 1c).

The collected samples were sent for analysis at the Key Laboratory of Groundwater Science and Engineering of Land and Resources, Institute of Hydrogeology and Environmental Geology, Chinese Academy of Geological Sciences. During sampling, the water temperature and pH were measured on site by a portable meter. Water samples used for indoor analysis were first filtered with 0.45 μm microporous filter membrane to remove suspended solids, and then loaded into three 100 mL polyethylene bottles that were washed twice with deionized water and dried; water samples needed to fill the sampling bottles to prevent gas from entering. We added 14 mol/L high-grade pure nitric acid to one of the water sample bottles until the pH value of the water sample was lower than 2.0. This water sample was used for routine cation and trace metal element analysis; no reagents were

added to the other two sample bottles for the determination of inorganic anions. Water samples for analysis of $\delta^2H$ and $\delta^{18}O$ were stored in brown bottles after filtration in the field. The main detection equipment for total analysis and trace elements were an ICP inductively coupled plasma emission spectrometer (ICAP6300) produced by Thermo Fisher company, and an ion chromatograph (Dionex ICS-1100) produced by Thermo Fisher company, USA. Standard samples were inserted at equal intervals during laboratory analysis to verify the accuracy of analysis results. The absolute error of all samples was not more than 3% by the error analysis of cationic and anion balance. The $\delta^2H$ and $\delta^{18}O$ isotopes were measured by a stable isotope ratio mass spectrometer (MAT-253) manufactured by Thermo Fisher, USA. The measured results were expressed as the thousandth difference compared with the Vienna Standard Mean Ocean Water (VSMOW) standard, and the accuracy was $\pm2.0$‰ and $\pm0.2$‰, respectively.

### 3.2. Previous Analysis Data

In order to better reveal the origin of carbonated spring water, this paper used previous analysis and analysis data of geothermal water chemistry, gas, and travertine. These included gas analysis data collected by the Qinghai New Energy Research Institute (1986) from the Yaoshuitan geothermal field, water quality analysis data from the Yaoshuitan geothermal field collected by Li et al. (2017) [28], travertine analysis data from Bingling Mountain in Xining Basin collected by Fu et al. (2019) [34], and groundwater gas analysis data collected from Qijiachuan, Pingan County by the China Railway First Survey and Design Institute Group Co., Ltd., Zheng et al. (2016) [32].

### 3.3. Research Methods

Firstly, the water source and $CO_2$ source of carbonated thermal water springs in Xining Basin were identified, and the water-bearing medium and surrounding rock of the formation and occurrence of carbonated thermal water springs were identified through the main components of anions in the carbonated thermal water springs. Then, the main water-rock interactions and chemical reactions during the formation of carbonated thermal water springs were determined by the composition of major anions and travertine deposited in the carbonated thermal water springs. Then, by analyzing the tectonic environment and geological conditions of carbon dioxide formation in the Earth's interior, combined with the tectonic environment of Xining Basin, the most likely formation conditions, formation process, migration path, and accumulation mode of $CO_2$ in carbonated thermal water springs were summarized, as well as the reason why geothermal springs were acidic. Finally, the temperature range of travertine formation and $CO_2$ generation was determined by the main mineral composition in the central calcium, and the thermal anomaly was determined by the temperature of deep heat storage, so as to determine whether the deep fault in the northern margin of Lajishan had a thermal control effect. To achieve these research objectives, the research methods used were as follows:

A Piper trilinear diagram and saturation index of main mineral components were used to determine the hydrochemical characteristics of geothermal water, carbonated thermal water springs, and groundwater in Xining Basin, and to determine the evolution path from recharge area to geothermal water. The molar concentration ratio of the main ion components in the water was used to analyze the hydrogeochemical process of the formation of the main ions in the carbonated thermal water spring and the mechanism of water-rock interaction. The relationship between $\delta^2H$ and $\delta^{18}O$ isotopes was used to determine the recharge source of geothermal water and carbonated thermal water springs in Xining Basin. Geothermal water $^{14}C$ dating and helium isotope R/Ra values were used to determine whether there may be a fluid (gas) source from the deep mantle in geothermal water. Calcareous uranium series dating was used to determine the formation age of geothermal water. The relationship between the $\delta^{13}C$ value of $CO_2$ and the volume fraction of $CO_2$ in carbonated thermal water springs was used to determine whether the $CO_2$ was of inorganic origin, biological origin, or both. The $\delta^{13}C$ range of travertine was used to

analyze whether the cause of travertine was surface travertine or endogenous travertine. The enrichment status of Eu (Europium) and Ce (Cerium) in travertine rare earth elements was used to determine the material source of travertine.

## 4. Results and Discussion

### 4.1. Geochemical Characteristics of Water, Gas, and Travertine

(1)  Hydrochemical characteristics of geothermal water (containing carbonated thermal water spring) and cold groundwater

The chemical composition of groundwater is the result of chemical reactions between water and minerals in the aquifer under certain temperatures and pressures. In a study of the causes of groundwater and the hydrogeological environment, it is necessary to scientifically classify the chemical composition of water. Piper proposed a trilinear diagram for the classification of water chemical composition in 1944, which was widely used later.

As can be seen from Figure 2, Zone I is the cold groundwater of Yaoshuitan in the front of the Laji Mountain, which is located in the front of the mountain and the thin overlying layer. The main hydrochemical types are $HCO_3 \cdot SO_4$-$Ca \cdot Na \cdot Mg$ or $HCO_3 \cdot SO_4$-$Ca \cdot Na \cdot Mg$, and the main anions are $HCO_3^-$ and $SO_4^{2-}$; the $HCO_3^-$ content is 312–345 mg/L. The mineralization degree is relatively low, 573–590 mg/L, which can represent the recharge area of groundwater in Xining Basin.

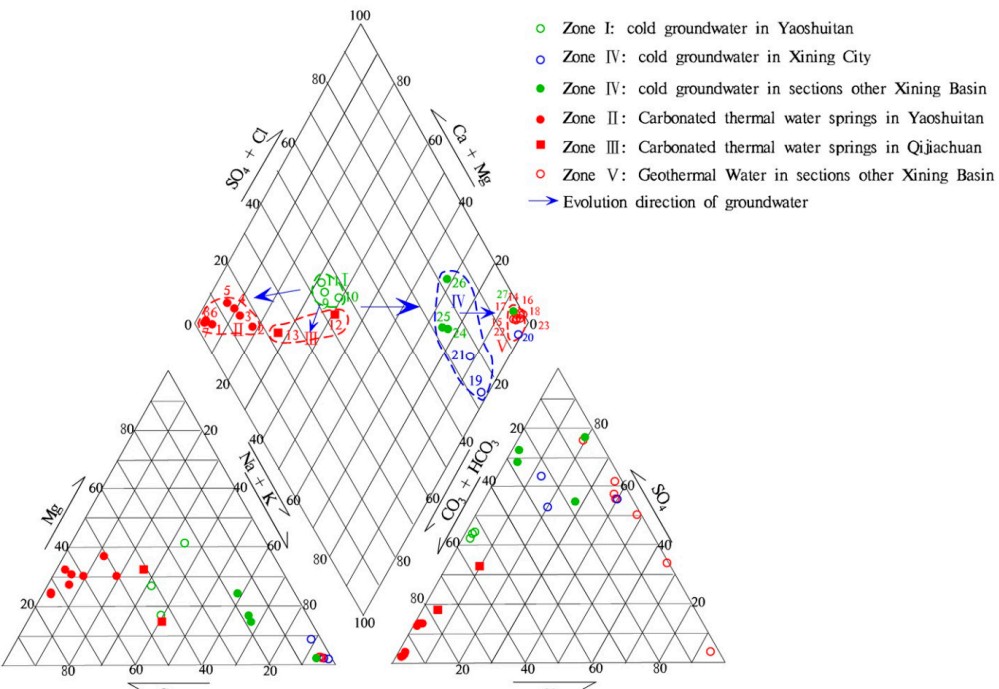

**Figure 2.** Piper trilinear diagram of geothermal water and cold groundwater in Xining Basin.

Zone II is the Yaoshuitan carbonated thermal water spring, the cations are mainly $Ca^{2+}$ and $Mg^{2+}$, and the hydrochemical types are mainly $HCO_3$-$Ca \cdot Mg$ and $HCO_3$-$Ca$ (Table 1). It shows remarkable characteristics of carbonate groundwater, and the salinity of geothermal water is 993–1850 mg/L, which is significantly higher than that of the cold groundwater below Shuitan beach. The main anion is $HCO_3^-$, and the content of $HCO_3^-$ is 996~1989 mg/L, which is significantly higher than that of cold groundwater in the same area. This could attribute to the mixing of a large amount of $CO_2$ from deep. The saturation index of soluble anhydrite and gypsum in Yaoshuitan carbonated thermal water spring is obviously higher than that of cold groundwater, and the saturation index of soluble stone salt is lower than that of cold groundwater, and the saturation index of relatively insoluble aragonite, calcite, and dolomite was higher than that of cold groundwater (Table 2 and

Figure 3). This is because the content of anhydrite, gypsum, aragonite, calcite, and dolomite in carbonate reservoir is relatively high, while the content of halite in cold groundwater aquifer of sandstone is relatively high.

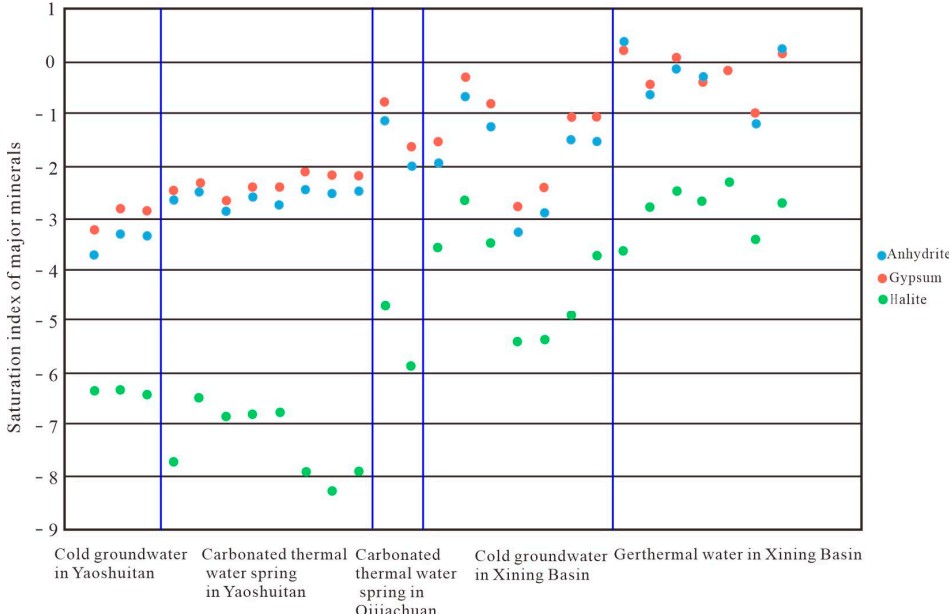

**Figure 3.** Saturation index distribution of main soluble minerals in carbonated thermal water spring, geothermal water, and cold groundwater in Xining Basin.

Zone III is the carbonated thermal water spring in the Qijiachuan carbonated thermal water spring in Pingan County. The mineralization degree of the Qijiachuan carbonated thermal water spring is 2940~5470 mg/L, which is obviously higher than that of the Yaoshuitan carbonated thermal water spring. The main anion is $HCO_3^-$, and the content of $HCO_3^-$ is 1820~3145 mg/L, which is obviously higher than that of cold groundwater in Xining Basin. The main hydrochemical types are $HCO_3 \cdot SO_4$-Mg·Na·Ca and $HCO_3 \cdot SO_4$-Ca·Na, which are more complex than Yaoshuitan carbonated thermal water spring. The saturation indices of anhydrite, gypsum, aragonite, calcite, dolomite, stone salt, and potassium salt are significantly higher than those of the Yaoshuitan carbonated thermal water spring and cold groundwater (Figure 3). It indicates that the runoff circulation process is relatively sufficient and there is obvious mixing between deep circulating carbonate geothermal water and shallow sandstone groundwater, and the degree of water-rock interaction is higher than that of the Yaoshuitan carbonated thermal water spring. In Figure 2, it is located on the right side of area II, and the location is relatively scattered. This is because the Qijiachuan is relatively far from the groundwater recharge area in the piedmont of the Laji Mountain, and there are thick sandstone and mudstone cap rocks above the carbonate reservoir. There is a phenomenon of frequent mixing of carbonate groundwater and sandstone groundwater.

Zone IV is the cold ground water in other parts of the Xining Basin, which is located in the runoff-discharge zone of shallow groundwater and is to the right on the Piper trilinear diagram (Figure 2). The salinity of groundwater ranges from 265 to 2820 mg/L. The salinity of shallow groundwater is equal to that of the recharge area. The salinity of cold groundwater significantly increases as it goes deeper into the basin. It indicates that, with the increase of water-rock interaction degree, the more soluble components of groundwater dissolved surrounding rock. The main hydrochemical types are $SO_4 \cdot Cl$-Na and $SO_4 \cdot HCO_3$-Na, which are more complex than the cold groundwater in the recharge area. The saturation indices of anhydrite, gypsum, aragonite, calcite, dolomite, stone salt, and potassium salt are significantly higher than those of the cold groundwater in the recharge area (Figure 3), indicating that the degree of water-rock interaction experienced was relatively high.

**Table 1.** Test results of main chemical components of carbonated thermal water springs, geothermal water, and groundwater in Xining Basin.

| Type | Sample Number | T (°C) | pH | K+ /mg/L | Na+ /mg/L | Ca2+ /mg/L | Mg2+ /mg/L | Cl− /mg/L | SO42− /mg/L | HCO3− /mg/L | δ18O/ VSMOW‰ | δ2H/ VSMOW‰ | TDS (mg/L) | Hydrochemical Type |
|---|---|---|---|---|---|---|---|---|---|---|---|---|---|---|
| Carbonated thermal water spring of Yaoshuitan | 1 | 37 | 6.68 | 4.63 | 20.1 | 317.4 | 93.7 | 7.5 | 45.4 | 1310.0 | | | 1850 | HCO3-Ca |
| | 2 | 41.5 | 6.91 | 2.50 | 97.5 | 225.4 | 83.2 | 12.7 | 153.7 | 1156.9 | −10.11 | −67.36 | 1550 | HCO3-Ca·Mg |
| | 3 | 35 | 7.12 | 3.81 | 47.9 | 196.7 | 88.1 | 6.3 | 129.5 | 995.9 | | | 993.5 | HCO3-Ca·Mg |
| | 4 | 38 | 7.03 | 3.75 | 47.4 | 285.7 | 88.4 | 9.1 | 153.0 | 1221.0 | | | 1221 | HCO3-Ca·Mg |
| | 5 | 22 | 7.05 | 2.36 | 43.1 | 394.7 | 97.1 | 7.3 | 176.9 | 1544.0 | | | 1510 | HCO3-Ca·Mg |
| | 6 * | 18.2 | 6.45 | 3.01 | 13.1 | 457 | 94.6 | 12.9 | 43.4 | 1774 | | | 1536 | HCO3-Ca·Mg |
| | 7 * | 21.5 | 6.12 | 1.86 | 9 | 309 | 88 | 12.9 | 27.1 | 1291 | −9.8 | −61 | 1111 | HCO3-Ca·Mg |
| | 8 * | 27.3 | 6.55 | 3.73 | 15.6 | 509 | 105 | 10.6 | 40.1 | 1989 | | | 1707 | HCO3-Ca·Mg |
| Cold groundwater of Yaoshuitan | 9 | 10 | 7.81 | 0.76 | 77.9 | 49.4 | 49.5 | 3.5 | 228 | 344.7 | | | 589.4 | HCO3·SO4-Mg·Na·Ca |
| | 10 | 7 | 7.74 | 1.13 | 89 | 83 | 20 | 5.25 | 206.5 | 312.3 | | | 574.1 | HCO3·SO4-Ca·Na |
| | 11 | 9 | 8.02 | 1.04 | 70 | 78.5 | 32.1 | 5.25 | 209.8 | 330 | | | 572.9 | HCO3·SO4-Ca·Na·Mg |
| carbonated thermal water spring of Qijiachuan | 12 | 21 | 6.54 | 0 | 756.5 | 739.5 | 145.4 | 294.2 | 1359 | 3145 | | | 5470 | HCO3·SO4-Ca·Na |
| | 13 | 17 | 6.63 | 68.8 | 189 | 300.5 | 145.5 | 58.2 | 320 | 1820 | −10.6 | −75 | 2940 | HCO3-Ca·Mg·Na |
| Geothermal water of Pingan | 14 | 67 | 7.20 | 80.69 | 15,100 | 549.6 | 176.1 | 22,600 | 1220 | 750 | | | 40,600 | Cl-Na |
| Geothermal water of Xining | 15 | 36 | 7.70 | 37.1 | 9352 | 242.6 | 116.9 | 5865 | 11,321 | 1427 | | | 27,696 | SO4·Cl-Na |
| | 16 | 34 | 7.44 | 42.9 | 14,460 | 548.4 | 139.8 | 11,729 | 15,370 | 524.5 | | | 42,604 | Cl·SO4-Na |
| | 17 | 61 | 7.56 | 124.0 | 11,100 | 270.1 | 78.3 | 6980 | 14,200 | 1410 | −10.4 | −74 | 34,200 | SO4·Cl-Na |
| | 18 | 53 | 8.41 | 48.74 | 16,600 | 425.6 | 140.0 | 9360 | 20,700 | 700 | −10.2 | −71 | 48,000 | SO4·Cl-Na |
| Cold groundwater of Xining | 19 | —— | 8.34 | 18.22 | 3650 | 41.36 | 19.36 | 1200 | 4160 | 2820 | | | 12,000 | SO4·Cl-Na |
| | 20 | 15.2 | 7.56 | 73 | 10,600 | 293.1 | 71.9 | 6860 | 12,900 | 1340 | | | 32,200 | SO4·Cl-Na |
| | 21 | 12.7 | 7.52 | 15 | 3092 | 280.6 | 63.9 | 899.7 | 5773 | 2720 | | | 12,300 | SO4·HCO3-Na |
| Other geothermal water in Xining Basin | 22 | 34 | 8.00 | 17.5 | 3735 | 141.2 | 32 | 1293 | 6258 | 508 | | | 11,992 | SO4·Cl-Na |
| | 23 | 62 | 7.65 | 37.3 | 14,330 | 567.7 | 107 | 15,993 | 10,726 | 249.8 | | | 41,820 | Cl·SO4-Na |
| Other cold groundwater in Xining Basin | 24 | 7 | 8.43 | 0.97 | 276.6 | 67.6 | 29.2 | 8.75 | 602 | 277 | | | 1139 | SO4·HCO3-Na |
| | 25 | 9 | 7.80 | 2.96 | 288 | 70.6 | 38.7 | 19.96 | 614 | 335.9 | | | 1226 | HCO3·SO4-Na |
| | 26 | 10 | 7.53 | 4.63 | 539.8 | 152.2 | 117.3 | 385.2 | 1021 | 429 | | | 2492 | SO4·Cl-Na·Mg |
| | 27 | 8 | 7.40 | 11.4 | 2312 | 120.5 | 18.4 | 745.5 | 4114 | 265.4 | | | 7590 | SO4-Na |

Note: Samples marked with an asterisk are from [28].

**Table 2.** Saturation index of main minerals in carbonated thermal water spring, geothermal water, and groundwater in Xining Basin.

| Type | Sample Number | T (°C) | pH | Anhydrite (CaSO₄) | Aragonite (CaCO₃) | Calcite (CaCO₃) | Dolomite CaMg(CO₃)₂ | Gypsum (CaSO₄·2H₂O) | Halite (NaCl) | Sylvite (KCl) | TDS | Hydrochemical Type |
|---|---|---|---|---|---|---|---|---|---|---|---|---|
| Carbonated thermal water spring of Yaoshuitan | 1 | 37 | 6.68 | −2.65 | 0.64 | 0.78 | 1.50 | −2.47 | −7.68 | −7.93 | 1850 | HCO₃-Ca |
| | 2 | 41.5 | 6.91 | −2.48 | 0.75 | 0.88 | 1.83 | −2.35 | −6.47 | −7.68 | 1550 | HCO₃-Ca·Mg |
| | 3 | 35 | 7.12 | −2.87 | 0.77 | 0.91 | 1.92 | −2.67 | −6.83 | −7.54 | 993.5 | HCO₃-Ca·Mg |
| | 4 | 38 | 7.03 | −2.58 | 0.93 | 1.07 | 2.10 | −2.42 | −6.78 | −7.50 | 1221 | HCO₃-Ca·Mg |
| | 5 | 22 | 7.05 | −2.74 | 0.94 | 1.09 | 1.88 | −2.41 | −6.76 | −7.57 | 1510 | HCO₃-Ca·Mg |
| | 6 * | 18.2 | 6.45 | −2.48 | 0.40 | 0.55 | 0.68 | −2.10 | −7.88 | −8.05 | 1536 | HCO₃-Ca·Mg |
| | 7 * | 21.5 | 6.12 | −2.53 | −0.14 | 0.01 | −0.21 | −2.19 | −8.24 | −8.47 | 1111 | HCO₃-Ca·Mg |
| | 8 * | 27.3 | 6.55 | −2.49 | 0.70 | 0.84 | 1.38 | −2.21 | −7.86 | −8.05 | 1707 | HCO₃-Ca·Mg |
| Cold groundwater of Yaoshuitan | 9 | 10 | 7.81 | −3.72 | 0.14 | 0.30 | 0.72 | −3.25 | −6.32 | −7.83 | 589.4 | HCO₃·SO₄-Mg·Na·Ca |
| | 10 | 7 | 7.74 | −3.32 | 0.22 | 0.37 | 0.19 | −2.82 | −6.30 | −7.68 | 574.1 | HCO₃·SO₄-Ca·Na |
| | 11 | 9 | 8.02 | −3.35 | 0.52 | 0.67 | 1.06 | −2.87 | −6.40 | −7.72 | 572.9 | HCO₃·SO₄-Ca·Na·Mg |
| Carbonated thermal water spring of Qijiachuan | 12 | 21 | 6.54 | −1.15 | 0.81 | 0.96 | 1.51 | −0.80 | −4.69 | / | 5470 | HCO₃·SO₄-Ca·Na |
| | 13 | 17 | 6.63 | −2.03 | 0.37 | 0.52 | 0.97 | −1.64 | −5.86 | −5.83 | 2940 | HCO₃-Ca·Mg·Na |
| Geothermal water of Pingan | 14 | 67 | 7.20 | 0.34 | 0.79 | 0.91 | 1.59 | 0.21 | −3.65 | −5.74 | 40,600 | Cl-Na |
| Geothermal water of Xining | 15 | 36 | 7.70 | −0.64 | 1.05 | 1.18 | 2.48 | −0.46 | −2.80 | −4.86 | 27,696 | SO₄·Cl-Na |
| | 16 | 34 | 7.44 | −0.15 | 0.62 | 0.76 | 1.33 | 0.04 | −2.49 | −4.69 | 42,604 | Cl·SO₄-Na |
| | 17 | 61 | 7.56 | −0.33 | 1.19 | 1.32 | 2.51 | −0.40 | −2.67 | −4.38 | 34,200 | SO₄·Cl-Na |
| | 18 | 53 | 8.41 | −0.19 | 1.57 | 1.69 | 3.36 | −0.19 | −2.32 | −4.60 | 48,000 | SO₄·Cl-Na |
| Cold groundwater of Xining | 19 | — | 8.34 | −1.94 | 1.09 | 1.24 | 2.40 | −1.54 | −3.56 | −5.39 | 12,000 | SO₄·Cl-Na |
| | 20 | 15.2 | 7.56 | −0.69 | 0.68 | 0.83 | 1.29 | −0.30 | −2.66 | −4.39 | 32,200 | SO₄·Cl-Na |
| | 21 | 12.7 | 7.52 | −1.25 | 1.09 | 1.25 | 2.03 | −0.82 | −3.49 | −5.32 | 12,300 | SO₄·HCO₃-Na |
| Other geothermal water in Xining Basin | 22 | 34 | 8.00 | −1.19 | 0.84 | 0.98 | 1.74 | −0.99 | −3.39 | −5.34 | 11,992 | SO₄·Cl-Na |
| | 23 | 62 | 7.65 | 0.23 | 0.74 | 0.86 | 1.34 | 0.14 | −2.70 | −5.08 | 41,820 | Cl·SO₄-Na |
| Other cold groundwater in Xining Basin | 24 | 7 | 8.43 | −3.28 | 0.69 | 0.84 | 1.39 | −2.78 | −5.37 | −7.30 | 1139 | SO₄·HCO₃-Na |
| | 25 | 9 | 7.80 | −2.91 | 0.20 | 0.36 | 0.55 | −2.43 | −5.35 | −6.83 | 1226 | HCO₃·SO₄-Na |
| | 26 | 10 | 7.53 | −1.52 | 0.24 | 0.40 | 0.81 | −1.05 | −4.90 | −6.47 | 2492 | SO₄·Cl-Na·Mg |
| | 27 | 8 | 7.40 | −1.55 | −0.36 | −0.21 | −1.13 | −1.06 | −3.72 | −5.53 | 7590 | SO₄-Na |

Note: Samples marked with an asterisk are from [28].

Zone V is the deep buried sandstone geothermal water in the urban area of Xining City and other parts of Xining Basin. The degree of water-rock interaction is high, which is on the right side of the Piper trilinear diagram (Figure 2). The mineralization degree of geothermal water ranges from 11,992 to 42,604 mg/L, which is obviously higher than that of the Yaoshuitan and Qijiachuan carbonated thermal water springs, and is also higher than that of cold groundwater in the basin. The main hydrochemical types are $SO_4 \cdot Cl$-Na and $Cl \cdot SO_4$-Na, which are more complex than the cold groundwater and geothermal water in the recharge area at the basin margin. The saturation indices of anhydrite, gypsum, aragonite, calcite, dolomite, stone salt, and potassium salt are significantly higher than those of carbonated thermal water springs in Yaoshuitan and Qijiachuan, and also significantly higher than those of cold groundwater in the basin (Figure 3). It shows that it has experienced a full runoff-cycle process and is in a closed hydrogeological environment inside the deeply buried sedimentary basin with slow runoff.

From the recharge area (Zone I) at the edge of Xining Basin to the cold groundwater area (Zone IV) and geothermal water area (Zone V) in the basin, it reflects the normal evolution path of the basin groundwater from the recharge area to the runoff and discharge area in the basin. From the recharge area of Xining Basin to the interior of the basin, the evolution of groundwater is mainly divided into two branches, one is from the recharge area at the basin edge to the cold groundwater area at the shallow part of the basin, and the other is from the recharge area at the basin edge to the geothermal water area in the deep part of the basin. From the recharge area at the edge of Xining Basin (Zone I) to the carbonated thermal water springs area near the edge of Xining Basin (Zone II and III), it is opposite to the normal evolution path of groundwater in the basin, which is due to the mixing of a large amount of $CO_2$ gas along the deep fault zone of the northern margin of the Laji Mountain along the edge of the basin. The addition of a large amount of $CO_2$ not only changes the hydrochemical type of groundwater, but also makes the anions $HCO_3^-$ and $SO_4^{2-}$, which change to $HCO_3^-$. It also makes the solubility of carbonated thermal water springs stronger and the degree of mineralization significantly increases. The pH values of carbonated thermal water spring range from 6.12 to 6.91, which is weakly acidic. Therefore, its genesis cannot be a strongly acidic magmatic heat source geothermal system [35].

(2) Water-Rock Interaction and Evolution Process

The ratio of $Na^+/Cl^-$ equal to 1 is usually attributed to salt dissolution. As shown in Figure 4a, the $Na^+/Cl^-$ ratio of the carbonated thermal water spring and its nearby groundwater is much higher than 1. Therefore, there is additional source of $Na^+$ in addition to halite dissolution. The molar ratio of $Na^+/Cl^-$ in the Qijiachuan carbonated thermal water spring with thick cap rock is higher than that in the Yaoshuitan carbonated thermal water spring, indicating that the deeper the carbonate reservoir is buried, the stronger the dissolution of silicate minerals from deep metamorphic sandstone and slate [36]. In addition, if $Ca^{2+}$, $Mg^{2+}$, $HCO_3^-$, and $SO_4^{2-}$ only come from calcite, dolomite, and gypsum weathering, the molar ratio of $(Ca^{2+} + Mg^{2+})$ and $(HCO_3^- + SO_4^{2-})$ should be 1. Figure 4b shows that the molar ratio of $(Ca^{2+} + Mg^{2+})$ and $(HCO_3^- + SO_4^{2-})$ is very close to 1, indicating that the water-rock interaction is dominated by the dissolution of calcite, dolomite, and gypsum in deep carbonate reservoirs. The $HCO_3^- + SO_4^{2-}$ value in the Qijiachuan carbonated thermal water spring is greater than the $Ca^{2+} + Mg^{2+}$ value, and the excess $HCO_3^- + SO_4^{2-}$ must be balanced by Na + produced by the dissolution of silicate minerals in metamorphic sandstone and slate. The molar ratio of $(Na^+ + K^+)/HCO_3^-$ in the Yaoshuitan carbonated thermal water spring, groundwater, and the Qijiachuan carbonated thermal water spring is far less than 1 (Figure 4c), and the molar ratio of $(Na^+ + K^+)/HCO_3^-$ in the Qijiachuan carbonated thermal water spring is smaller than in Yaoshuitan. It indicates that the dissolution of silicate minerals plays a minor role in the chemical composition formation of carbonated thermal water springs in Xining Basin. These results indicate that the hydrochemical composition of Yaoshuitan and Qijiachuan carbonated thermal water springs in Xining Basin is likely controlled by the dissolution of calcite, dolomite, and gypsum in deep carbonate reservoirs, supplemented by the dissolution of silicate minerals. The deeper the

burial of carbonate reservoirs, the thicker the sandstone and mudstone cap rocks, and the stronger the effect of dissolution of silicate minerals on chemical composition [37,38].

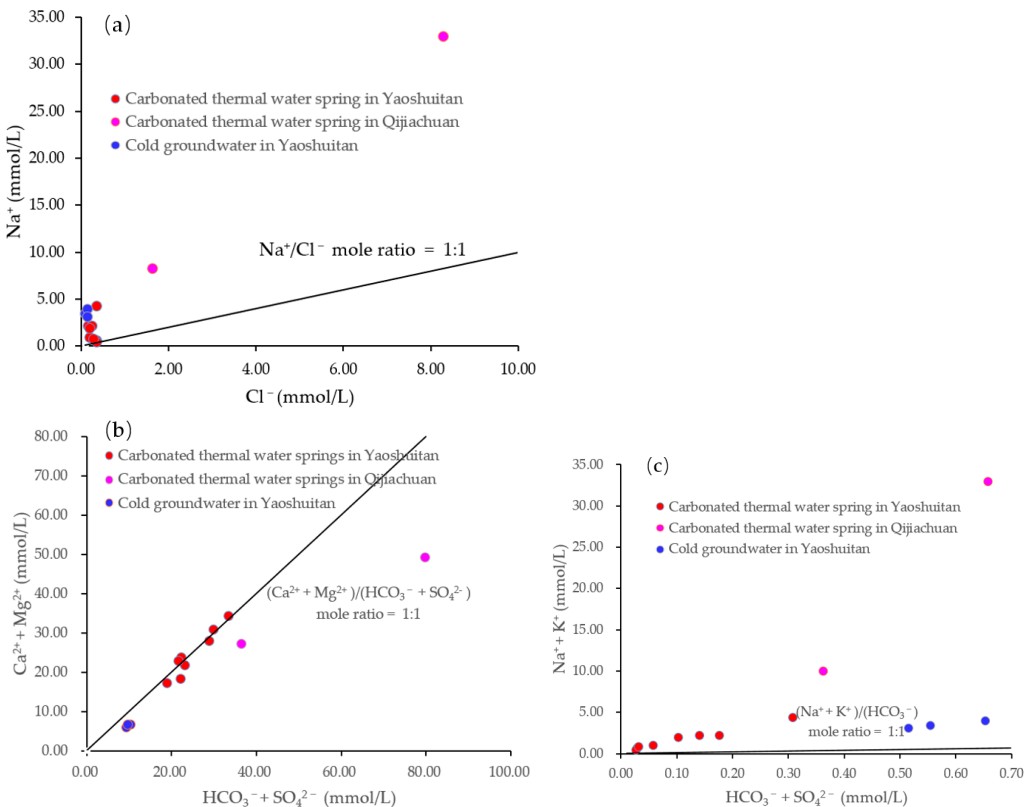

**Figure 4.** (**a**) Relationship between $Na^+$ and $Cl^-$, (**b**) relationship between $Ca^{2+} + Mg^{2+}$ and $HCO_3^- + SO_4^{2-}$, (**c**) relationship between $Na^+ + K^+$ and $HCO_3^-$ for water samples from Yaoshuitan and Qijiachuan.

(3)    Gas composition and weak acid origin

The gas sample compositions of the Yaoshuitan and Qijiachuan carbonated thermal water springs show that the gas component in the carbonated thermal water springs is mainly $CO_2$, and $CO_2$ in most sample gasses is about 90%, followed by $N_2$ and $O_2$ (Table 3). The $CO_2$ volume fraction of the Yaoshuitan geothermal field is slightly lower than that of the Qijiachuan geothermal field, which may be due to the thin cover layer of the Yaoshuitan geothermal field, and the more adequate contact between the geothermal water and the atmosphere, thus absorbing more air components. The Qijiachuan geothermal field has thick cap rocks, which isolate the contact between the geothermal water and the air, so the $CO_2$ volume fraction is high.

**Table 3.** Composition and content of gas phases in groundwater studied.

| Number | Sampling Location | Gas Sample Composition Data (%) | | | | | PDB/$\delta^{13}C_{CO2}$ Accuracy: ±(0.2~0.5)‰ | $^3He/^4He$ | R/Ra |
|---|---|---|---|---|---|---|---|---|---|
| | | $CO_2$ | $N_2$ | $O_2$ | Ar | $CH_4$ | | | |
| 1 [#] | Shangraozhuang | 90.28 | 7.45 | 2.17 | 0.10 | 0 | −1.94 | $1.42 \times 10^{-6}$ | 1.01 |
| 2 [#] | Binglingshan 1 | 93.95 | 4.59 | 1.40 | 0.067 | 0 | −3.27~−2.24 | $1.02 \times 10^{-6}$ | 0.73 |
| 3 [#] | Binglingshan 2 | 94.14 | 4.44 | 1.35 | 0.067 | 0 | −2.53 | $0.62 \times 10^{-6}$ | 0.44 |
| 4 [#] | ZK10 well | 87.63 | 9.66 | 2.58 | 0.13 | 0 | −1.46 | $0.0626 \times 10^{-6}$ | 0.04 |
| 5 [*] | Yaoshuitan 16 | 64.12 | 31.03 | 4.44 | 0.36 | 0 | —— | —— | —— |
| 6 [*] | Yaoshuitan 64 | 88.12 | 10.36 | 1.34 | 0.12 | 0 | —— | —— | —— |

Notes: Samples marked with "#" are from [32]; samples marked with "*" are from the internal data of the Qinghai New Energy Research Institute (1986).

There are generally three sources of $CO_2$ in groundwater: (1) Soil $CO_2$ produced by the decomposition of organic matter (humus) in the soil or the respiration of plant roots. The $\delta^{13}C$

value of $CO_2$ produced by different climate zones is different. The $\delta^{13}C$ value of $CO_2$ produced by vegetation in warm climate zones is $-25‰$, and the average in semi-arid areas is $-15‰$. (2) Atmospheric $CO_2$ dissolved in rainwater. Atmospheric $CO_2$ partial pressure is small and can be ignored in most cases, but in areas with little or no vegetation, atmospheric $CO_2$ may account for a large proportion. The $\delta^{13}C$ value of atmospheric $CO_2$ is about $-7‰\sim-9‰$. (3) $CO_2$ originating from volcanic or magmatic activity. The $\delta^{13}C$ value of $CO_2$ from this deep geothermal source varies between $-2$ and $-6‰$. The $\delta^{13}C$ of mantle-derived $CO_2$ is $-4‰\sim-11‰$. The $\delta^{13}C$ of metamorphic $CO_2$ of limestone is $\pm3‰$ [39].

According to the relationship between the $CO_2$ volume fraction and the $\delta^{13}C$ value (PDB) of the carbon isotope of $CO_2$ in the Qijiachuan carbonated thermal water spring (Figure 5, Table 3), the $\delta^{13}C$ value is $-3.27\sim1.46‰$, and its $CO_2$ sources are consistent, all of which are of inorganic origin, formed by the metamorphic decomposition of carbonate and marble in the deep basement. Since calcite is the main component of travertine, according to Muffler and White (1969), the reaction temperature is mainly $150\sim200$ °C [40]. The chemical reaction formula is as follows:

$$CaCO_3 \overset{150\sim200°C}{\rightarrow} CaO + CO_2 \uparrow \tag{1}$$

$$CaMg(CO_3)_3 \overset{150\sim200°C}{\rightarrow} CaO + Mg + 2CO_2 \uparrow \tag{2}$$

$$CaCO_3 + SiO_2 \overset{150\sim200°C}{\rightarrow} CaSiO_3 + CO_2 \uparrow \tag{3}$$

$$CaMg(CO_3)_3 + SiO_2 \overset{150\sim200°C}{\rightarrow} CaMgSiO_{4-} + 2CO_2 \uparrow \tag{4}$$

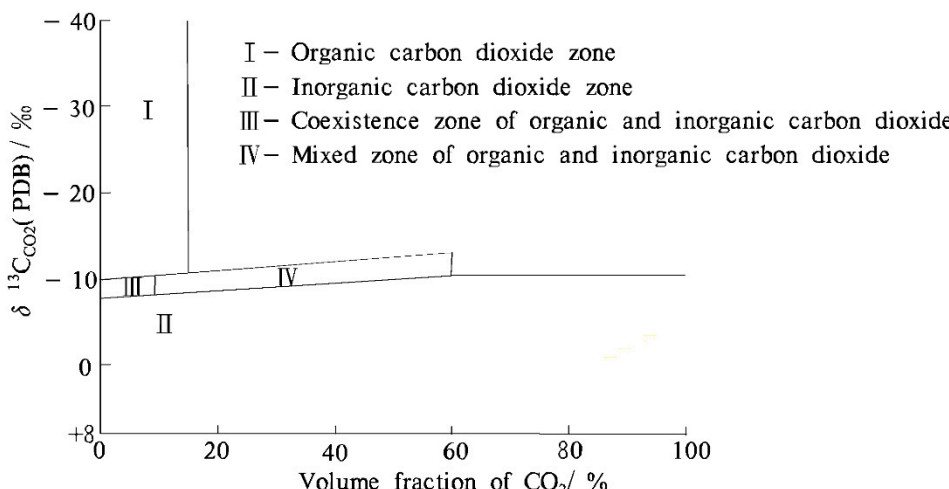

**Figure 5.** Identification chart of carbon dioxide origin in Xining Basin (according to [41]).

Muffler and White (1968) studied the $CO_2$ generated by the mineralization of carbonate rocks and silicate rocks in the Salton Sea geothermal field in southeastern California, USA, and found that with an increase in temperature and pressure, the mineral composition would undergo different metamorphism. When the temperature is lower than 150 °C, the content of dolomite and kaolinite in the clastic sediments is abundant; when the temperature is 150~200 °C, the dolomite and kaolinite decompose to form chlorite, calcite, and $CO_2$. When the temperature exceeds 300~320 °C, calcite gradually decreases, forming epidote and $CO_2$ [40]. The equation is as follows:

$$\underset{\text{Dolomite}}{4MgCa(CO_3)_2} + \underset{\text{kaolinite}}{2.5Al_2Si_2O_2(OH)_4} + \underset{\text{ankerite}}{5CaMg_{0.6}Fe_{0.4}(CO_3)_2} + H_2O + Fe^{3+} + 1.5O^{2-} \rightarrow \underset{\text{chlorite}}{Mg_7Al_2Fe_2^{2+}Fe^{3+}Si_5Al_3O_{20}(OH)_{16}} + \underset{\text{calcite}}{9CaCO_3} + \underset{\text{carbond ioxide}}{9CO_2} \tag{5}$$

$$\underset{\text{Muscovite}}{K_2Al_4Si_6Al_2O_{20}(OH)_4} + \underset{\text{calcite}}{4CaCO_3} + \underset{\text{quartz}}{6SiO_2} + 2Fe^{3+} + 3O^{2-} \rightarrow \underset{\text{epidote}}{2Ca_2Al_2Fe^{3+}Si_3O_{12}(OH)} + \underset{\text{K-feldspar}}{2KAlSi_3O_8} + \underset{\text{carbon dioxide}}{4CO_2} + H_2O \tag{6}$$

Calcite is the main component in carbonated thermal water spring travertine at Qijiachuan, accounting for 66 to 97% [42], and the pores of the travertine are mostly filled with secondary calcite [34], which indicates that the carbonated thermal water spring travertine in the Qijiachuan may be formed in the deep geological environment with a temperature of 150–200 °C. The $CO_2$ formed during the metamorphism of deep carbonates and silicates under high temperature and pressure migrates along the surface of the Laji Mountain fault zone and its secondary faults or the faults where it intersects, forming weakly acidic carbonated thermal water spring [43]. Weak acidity enhances the solubility of water to carbonate. This carbonated thermal water spring comes into contact with carbonate rock to further generate $CO_2$ and form calcification at the discharge point. At the same time, because the dissolution reaction of acid water to carbonate rocks consumes $H^+$, the acidity of water becomes weak, so the carbonated thermal water spring shows weak acidity, rather than strong acidity. The general reaction is as follows:

$$CO_2 + H_2O \rightarrow H_2CO_3 \rightarrow H^+ + HCO_3^- \tag{7}$$

$$Ca^{2+} + 2HCO_3^- \rightarrow CaCO_3 + H_2O + CO_2\uparrow \tag{8}$$

$$CaCO_3 + 2H^+ \rightarrow Ca^{2+} + CO_2\uparrow + H_2O \tag{9}$$

(4)     Isotopic composition of water

In order to describe the linear relationship between $\delta^2H$ and $\delta^{18}O$ stable isotopes in Meteoric Water, Craig published the famous Global Meteoric Water Line (GMWL) equation in 1961 [44]. The Craig line should be widely used as a comparative benchmark in the study of water cycles, especially precipitation processes [45]. As can be seen from Figure 6, geothermal water in the study area falls near the GMWL line, indicating that the main recharge source of geothermal water in the study area is atmospheric precipitation. However, the $^{18}O$ of the Yaoshuitan carbonated thermal water spring has an obvious negative drift, which is because the geothermal spring contains a large amount of $C^{16}O_2$ gas. The $^{16}O$ in $C^{16}O_2$ reacts with the $^{18}O$ in water by displacement (equilibrium fractionation). The reaction formula is as follows:

$$C^{16}O_2 + H_2{}^{18}O = C^{18}O^{16}O + H_2{}^{16}O \tag{10}$$

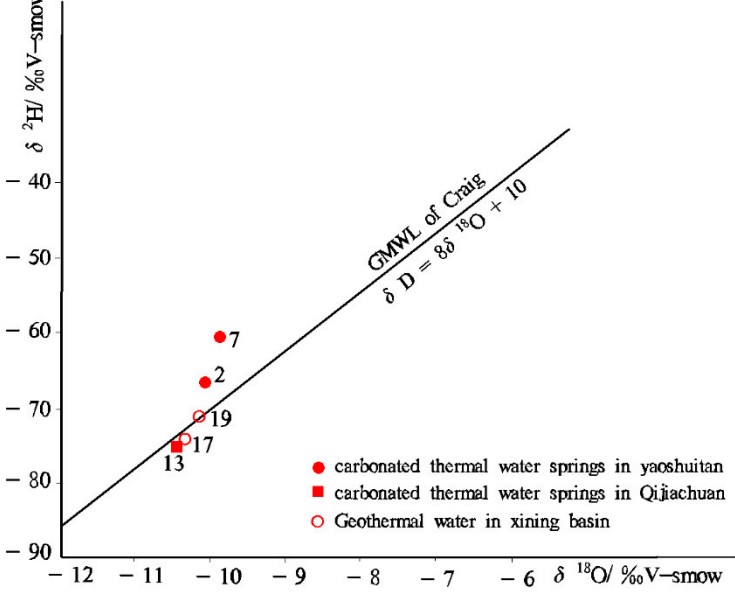

**Figure 6.** The relationship between $\delta^2H$ and $\delta^{18}O$ in the geothermal water of Xining Basin.

Displacement reaction caused $^{18}O$ in the carbonated thermal water spring to decrease and $^{18}O$ negative drift occurred [46].

Sun (2015) and Li et al. (2007) conducted $^{14}C$ dating of geothermal water in the Yaoshuitan and Qijiachuan carbonated thermal water springs, and showed that the $^{14}C$ age of carbonated thermal water spring was 25.50 ± 0.50~47.61 ± 2.86 ka [28,47], which was similar to the age of sandstone geothermal water in Xining Basin. The presence of more than 20,000-year-old components in the shallow carbonated thermal water spring in the recharge area indicates that some of these carbonated thermal water spring may come from deep fluid (gas) sources. Fu et al. (2019) conducted uranium-series dating of carbonated thermal water spring travertines at Qijiachuan, showing that the travertine ages approximately range from 16.82 ± 1.38 to 53.768 ± 1.63 ka (Table 4), which is roughly equivalent to the age of local carbonate geothermal water [34].

**Table 4.** Dating analysis results of travertine uranium series of carbonated thermal water springs in Qijiachuan, Ping'an County (according to [34]).

| Number | U (μg/g) | Th (μg/g) | $U^{234}/U^{238}$ | $Th^{230}/Th^{232}$ | Age/ka |
|---|---|---|---|---|---|
| 1 | 22.0 | 0.338 | 2.37 | 2.44 | 24.459 ± 2.17 |
| 2 | 9.42 | 3.57 | 1.09 | 2.10 | 16.820 ± 1.38 |
| 3 | 19.2 | 0.23 | 2.47 | 1.93 | 28.306 ± 2.84 |
| 4 | 19.4 | 0.718 | 1.53 | 5.50 | 23.130 ± 1.96 |
| 5 | 10.9 | 2.08 | 1.46 | 1.48 | 43.167 ± 5.26 |
| 6 | 28.1 | 2.37 | 1.46 | 0.65 | 23.973 ± 1.66 |
| 7 | 10.0 | 3.74 | 1.01 | 1.55 | 53.768 ± 1.63 |

It can be seen from Table 3 that the R/Ra of the carbonated thermal water spring samples at Qijiachuan is mostly less than 1, indicating that the helium in the geothermal water is mainly of crust-derived helium origin, and there is basically no deep mantle-derived material source.

(5)    Travertine characteristics

Pentecost (1995) found that thermogenic travertines mostly occur in areas with strong neotectonic activity, where there is high $CO_2$ emission [48]. The $\delta^{13}C$ range of carbonated thermal water spring travertine in Qijiachuang River is +10.57‰~+11.99‰ [49], and the $\delta^{13}C$ value of endogenous travertine is in the range of −2‰~+10‰ [50], indicating that it is due to the degasification of thermogenic $CO_2$.

The content characteristics of rare earth elements in groundwater are usually related to the pH value and redox conditions of the geological environment. The pH value is the most important physical and chemical parameter affecting the content of rare earth elements in geological environments. The chemical properties of Eu (Europium) and Ce (Cerium) in rare earth elements are active, and the reaction to the acidity and alkalinity of geological environment is very obvious, which is prone to fractionation during whole geological process. Under the condition of acidic medium, Eu can easily migrate with the medium, so the content decreases, and Ce is enriched, while under the condition of alkaline medium, the chemical properties of Ce are active, and the content decreases with the migration of the medium, and, at the same time, Eu is enriched [51]. The travertine analysis data from Qijiachuan showed that δEu was significantly less than 1, indicating that Eu was in a state of deficit, and δCe was greater than 1, indicating that Ce was enriched [34]. It indicates that the groundwater providing the source of travertine materials has been under acidic conditions for a long time, which is supplied by $CO_2$ from deep in the Earth's crust. This kind of groundwater was degassed and deposited to form travertine when it reached the surface. This slightly acidic environment ensures the continuous reaction of carbonated thermal water springs and carbonate rocks, as well as the continuous generation of $CO_2$, resulting in the release of a large amount of $CO_2$ from surface discharge outlets.

*4.2. Genesis of Carbonated Thermal Water Spring and Geothermal Significance*

The $\delta^2$H and $\delta^{18}$O isotopic characteristics indicate that the recharge source of carbonated thermal water springs in Xining Basin is mainly atmospheric precipitation. The $^{14}$C ages of carbonated thermal water springs are $25.50 \pm 0.50 \sim 47.61 \pm 2.86$ ka, indicating that some of these carbonated thermal water springs may be derived from deep fluid (gas) sources. The R/Ra in the carbonic acid geothermal spring is mostly less than 1, indicating that the helium in the geothermal water is mainly caused by crust-derived helium, and there is basically no deep mantle-derived material source.

The Piper trilinear diagram shows that from the recharge area at the edge of Xining Basin to the Yaoshuitan and Qijiachuan carbonated thermal water spring areas, the direction of the normal groundwater evolution path of the basin is contrary to that of the basin, which is due to the large amount of $CO_2$ gas mixed in the deep fault along the northern margin of Laji Mountain. The ratio of ($Ca^{2+} + Mg^{2+}$) and ($HCO_3^- + SO_4^{2-}$) in the Yaoshuitan carbonated thermal water spring is close to 1, indicating that the water-rock interaction is dominated by the dissolution of calcite, dolomite, and gypsum in deep carbonate reservoir. The molar concentration of $HCO_3^- + SO_4^{2-}$ in the Qijiachuan carbonated thermal water spring is significantly higher than that of $Ca^{2+} + Mg^{2+}$. The molar ratio of ($Na^+ + K^+$)/$HCO_3^-$ in the Yaoshuitan carbonated thermal water spring, groundwater, and the Qijiachuan carbonated thermal water spring is far less than 1. It is indicated that the hydrochemical composition of the Yaoshuitan and Qijiachuan carbonated thermal water springs in Xining Basin is likely controlled by the dissolution of calcite, dolomite, and gypsum in deep carbonate reservoirs, supplemented by the dissolution of silicate minerals. The $\delta^{13}$C value of the Qijiachuan carbonated thermal water spring is $-3.27 \sim -1.46‰$. The relationship between $CO_2$ volume fraction and $\delta^{13}$C value (PDB) of carbon isotopes of $CO_2$ indicates that the $CO_2$ source is of inorganic origin, which is mainly formed by metamorphic decomposition of deep carbonate and marble. Calcite is the main component of carbonated thermal water spring travertine, indicating that the reaction temperature is mainly 150~200 °C. The $\delta$Eu < 1 and $\delta$Ce > 1 in the calcium center of carbonated thermal water springs indicate that the groundwater supplying the travertine material has been in the acidic environment receiving the $CO_2$ supply from the deep crust for a long time. The groundwater reaches the surface and degasses and deposits to form travertine.

The $CO_2$ formed during the metamorphism of deep carbonate and silicate under high temperature and high pressure migrates along the Lajishan fault zone and its secondary faults or the surface of the faults that intersect with it, forming weak acid carbonated thermal water springs. Acidity enhances the solubility of water to carbonate rocks, and this carbonated thermal water spring comes into contact with carbonate rock to further generate $CO_2$. At the same time, due to the dissolution reaction of acidic water to carbonate rocks, $H^+$ is consumed, which weakens the acidity of water and makes carbonated thermal water springs weakly acidic.

The genetic mechanism of the carbonated thermal water springs at Yaoshuitan and Qijiachuan in Xining Basin can be summarized as follows: A series of tectonic activities, such as the late collision and post-collision between India and the Eurasian plate, led to the uplift of the northern Qinghai–Tibet Plateau and the asthenosphere upwelling, thermal intrusion, and other deep dynamic processes, which kept a higher thermal background in the eastern Qinghai Province, including Xining Basin. The heat source of carbonated thermal water springs comes from the lower crust and upper mantle. The deep faults on the northern margin of Laji Mountain with obvious neotectonic activity and other deep and large faults provide channels for the intrusion of deep crust–mantle thermal materials, and local geothermal anomalies are formed near the deep and large faults. Concealed carbonate rocks and silicate rocks with greater thicknesses provide the material basis for $CO_2$ generation. Carbonates and silicates undergo thermal metamorphism under the high-temperature and high-pressure environment in the deep geothermal anomaly to form a large amount of $CO_2$. $CO_2$ dissolved in water enhances the acidity of water, it migrates to the surface along the Laji Mountain fault zone and its secondary faults or the faults

that intersect with it, and dissolves in the shallow groundwater to form weakly acidic carbonated thermal water spring. The excretion site forms travertine. This carbonated thermal water spring comes into contact with the carbonate rock to further generate $CO_2$, and, at the same time, forms travertine at the discharge point (Figure 7). Because the dissolution reaction of acid water to carbonate rocks consumes $H^+$, the acidity of water becomes weak, so the carbonated thermal water springs show weak acidity, rather than strong acidity.

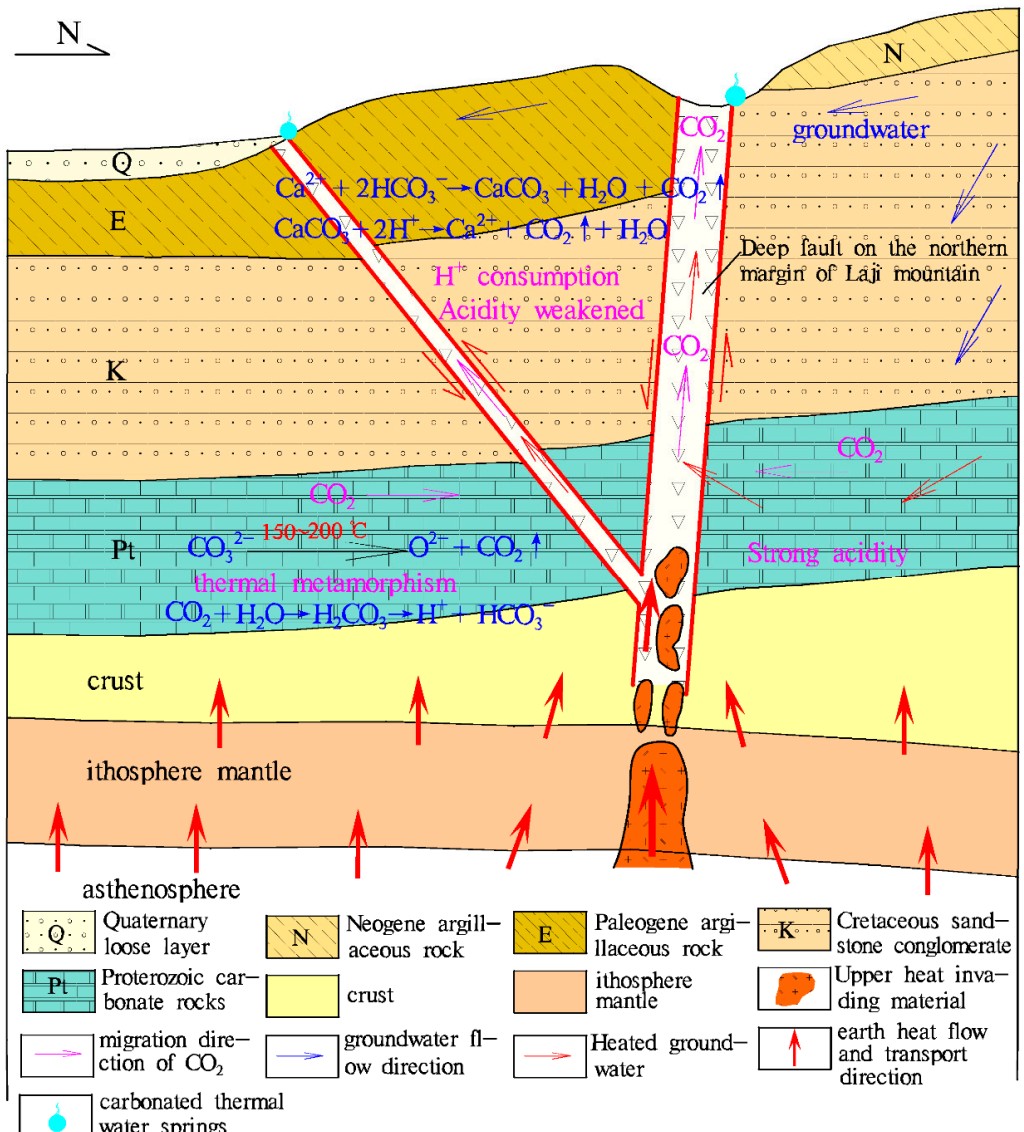

**Figure 7.** Typical $CO_2$ migration and accumulation model.

The carbonated thermal water spring travertine mainly comprises calcite, indicating that travertine may be formed in a deep geological environment with a temperature of 150~200 °C. Calculated according to the maximum burial depth of regional carbonate rock stratum of 3000~4000 m, it is impossible to reach such a high temperature in the carbonate strata under the normal geothermal gradient (3 °C/100 m), which indicates that there is an abnormal heat source in the deep carbonate strata. This anomalous heat source may be caused by the deep fault in the northern margin of Laji Mountain, and other deep and large faults conducting a deep crust–mantle heat source, which indicates that the deep fault in the northern margin of Laji Mountain has an obvious heat-controlling effect, and it also

offers good exploratory potential of geothermal resources near the deep and large fault at the edge of Xining Basin.

## 5. Conclusions

(1)  The $\delta^2H$ and $\delta^{18}O$ isotopic characteristics indicate that the recharge source of carbonated thermal water springs in Xining Basin is mainly atmospheric precipitation. The $^{14}C$ age of carbonated thermal water springs is more than 20 ka, indicating that some of these carbonated thermal water springs may come from deep fluid (gas) sources. The R/Ra in the carbonated thermal water spring is mostly less than 1, indicating that the helium in the geothermal water is mainly caused by crust-derived helium, and there is basically no deep mantle-derived material source.

(2)  The Piper trilinear diagram shows that from the recharge area at the edge of Xining Basin to the Yaoshuitan and Qijiachuan carbonated thermal water spring areas, the direction of the normal groundwater evolution path of the basin is contrary to that of the basin, which is due to the large amount of $CO_2$ gas mixed in the deep fault along the northern margin of Laji Mountain. The ratios of $(Ca^{2+} + Mg^{2+})$ and $(HCO_3^- + SO_4^{2-})$ and f $(Na^+ + K^+)/HCO_3^-$ in the Yaoshuitan and Qijiachuan carbonated thermal water springs indicate that the hydrochemical composition of the Yaoshuitan and Qijiachuan carbonated thermal water springs in Xining Basin is controlled by the dissolution of calcite, dolomite, and gypsum in deep carbonate reservoirs, supplemented by the dissolution of silicate minerals. The relationship between $CO_2$ volume fraction and $\delta^{13}C$ value (PDB) of carbon isotopes of $CO_2$ indicates that the $CO_2$ source is of inorganic origin, which is mainly formed by metamorphic decomposition of deep carbonate and marble. The $\delta Eu < 1$ and $\delta Ce > 1$ in the calcium center of carbonated thermal water springs indicate that the groundwater supplying the traverite material has been in the acidic environment receiving the $CO_2$ supply from the deep crust for a long time.

(3)  A series of tectonic activities, such as late collision and post-collision between the Indian and Eurasian plates, led to the uplift, asthenosphere upwelling, and thermal invasion of the northern Tibetan Plateau and other deep dynamic processes. The deep faults in the northern margin of the Laji Mountain and other deep faults with obvious neotectonic activity provided channels for the up-invasion of deep thermal materials, and local geothermal anomalies were formed near the deep faults. The hidden carbonate rocks and silicate rocks with large thickness undergo thermal metamorphism under high temperature and high pressure in the deep geothermal anomaly area and form a large amount of $CO_2$, which is dissolved in water and enhances the acidity of water. At the same time, the dissolution reaction of acidic water to carbonate rocks consumes $H^+$, which keeps the carbonated thermal water springs weakly acidic.

(4)  The composition of travertine in carbonated thermal water springs is dominated by calcite, indicating that travertine may be formed in a deep geological environment with a temperature of 150~200 °C, indicating that there are abnormal heat sources in shallow carbonate strata with a burial depth of 3000~4000 m. The abnormal heat source may be caused by the deep fault in the northern margin of Laji Mountain and other deep and large faults channeled the deep crust and mantle heat source, indicating that the deep fault in the northern margin of Laji Mountain has an obvious heat-controlling effect, and there is a good prospect of geothermal resources exploration near the fault.

**Author Contributions:** Writing—original draft preparation, Y.L. (Yude Lei), Z.Z. and B.Z.; Conceptualization, writing, review and editing, B.Z.; thesis writing instructor, G.W.; English writing and drawing of the paper, J.G.; field investigation, X.T. and D.Z.; field investigation and sampling, Y.L. (Yude Lei), Z.Z. and Y.L. (Yinfei Luo). All authors have read and agreed to the published version of the manuscript.

**Funding:** This work was jointly funded by the Applied Basic Research Project of Qinghai Provincial Department of Science and Technology (2019-ZJ-7062), Self-funded Science and Technology Project of Qinghai 906 Engineering Investigation and Design Institute (2021-KJ-002), and China Geological Survey Project (DD20189114 and DD20190132).

**Data Availability Statement:** The raw data supporting the conclusions of this article will be made available by the authors, without undue reservation.

**Conflicts of Interest:** The authors declare no conflict of interest.

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
