# Peer review of "Genesis of Significance of Carbonated Thermal Water Springs in Xining Basin, China"

_water, doi:10.3390/w14244058_

Round 1

Reviewer 1 Report

Manuscript ID: water-1946394
Title: Genesis of Carbonated Hot Springs in Xining Basin and Its Geothermal Significance.

The manuscript deals with the geochemistry of groundwaters in the Xining Basin. Both, geothermal and cold waters are presented. Additionally isotopic composition of carbon of  travertine is compared with isotopic composition of carbon in DIC.

The manuscript contains a lot of shortages, especially in adequate presentation of provided geochemical data. Description of data and their geochemical interpretation is very generic and not sufficiently clear; cold and hot water are confused in interpretation of their origin; relations between these two group of waters is unclear. There presentation of chemical analysis of geothermal waters and cold waters is chaotic and unclear. The conclusions regarding the origin of geothermal springs are based only on the interpretation of local and regional geology and tectonic processes and have hardly something in common with the geochemical data presented and generally described in the manuscript. This strongly suggest that the conclusions are not sufficiently supported by the provided geochemical data – first of all due to very generic and, in some places, not adequate (i.e. not sufficiently justified) interpretation of the data.

 That is why I have to recommend the manuscript for rejection and resubmission after serious improvements of interpretation of geochemical data.

My major detailed remarks are attached in the pdf file.

Author Response

Dear reviewer,

Thank you for your valuable comments, which can significantly improve the quality of this article. First of all, I would like to declare that the use of geochemical characteristics of geothermal water, gas composition, etc., is a relatively mature method to determine the origin of carbonated hot springs. The main purpose of this paper is to judge that the carbon dioxide in the carbonate hot spring in the study area is formed by thermal metamorphism in the carbonate thermal reservoir. The heat source is intruded along the deep fault of the northern margin of Lajishan, and it is judged that the Lajishan fault is a thermal control fault, which has a good prospect of geothermal resources exploration. Previous studies have not yet clearly understood the thermal control effect of the Lajishan fault. The results of this study confirm that the Lajishan fault is a thermal control fault. In addition, this paper supplements the ' evolution of H + content from deep to shallow during the formation of carbonate hot springs ', which is also the innovation of this paper. See the attachment for details.

Sincerely,

Baojian Zhang

Reviewer 2 Report

This is original paper but the role of pH is not mention satisfactorily. Especially, the writers should add something in Fig. 5 and explaining that how pH changes spatially and temporally in the proposed model. pH changes from deep dissolution of limestone through path way to the surface. There are some small items and I wrote my suggestions on the text. 

Author Response

Dear reviewer,

Thank you for your valuable comments, which can significantly improve the quality of this article. In the paper and Fig. 6, the evolution characteristics of H+ content from deep to shallow are added. CO2 from deep thermal metamorphism is dissolved in water to form acid water, which enhances the solubility of water to carbonate rocks. This carbonate spring and carbonate rocks are translated into CO2. At the same time, because the acid water consumes H+ in the dissolution reaction of carbonate rocks, the acidity of water becomes weak, so that the carbonate hot spring is weakly acidic instead of strongly acidic.

Sincerely,

Baojian Zhang

Reviewer 3 Report

I read carefully the revised manuscript number:water-1946394, the revised manuscript entitled: "Genesis of carbonated hot springs in Xining Basin and its geothermal significanc".

Unfortunately, the authors either did not respond to the reviewers' comments at all or responded in part, and the manuscript was not corrected.Check the English Grammar. The English language is moderate. Please check all parts of the manuscript and correct grammatical errors. The authors should ask the help of native English speaking proofreader, because there are some linguistic mistakes that should be fixed. Nevertheless, the manuscript is not acceptable in its current form. Since the quality of the study is lower than average for the publication in the journal, tools for objective function optimization are unclear in the methodology and conclusion, but it still needs a major revisions before reconsideration. I attached my reviewer comments in the PDF file. Authors should apply all of my comments.

Author Response

Dear reviewer,

Thank you for your valuable comments, which can significantly improve the quality of this article. First of all, I would like to declare that the use of geochemical characteristics of geothermal water, gas composition, etc., is a relatively mature method to determine the origin of carbonated hot springs. The main purpose of this paper is to judge that the carbon dioxide in the carbonate hot spring in the study area is formed by thermal metamorphism in the carbonate thermal reservoir. The heat source is intruded along the deep fault of the northern margin of Lajishan, and it is judged that the Lajishan fault is a thermal control fault, which has a good prospect of geothermal resources exploration. Previous studies have not yet clearly understood the thermal control effect of the Lajishan fault. The results of this study confirm that the Lajishan fault is a thermal control fault. In addition, this paper supplements the ' evolution of H + content from deep to shallow during the formation of carbonate hot springs ', which is also the innovation of this paper. See the appendix for details.

Sincerely,

Baojian Zhang

Round 2

Reviewer 1 Report

After second review of this manuscript I am satisfied with major improvements made by the authors in relation to provided new or additional content.

However there is still many important technical shortcomings connected with the use of professional terminology and the use of English language which needs extensive improvements. In my opinion all these terminological mistakes should be carefully corrected by the authors; actually it is lower than the average for this kind of papers published in Water journal. Moreover, some sections, first of all Section 3 (methods & analysis) needs further extensive improvements.

That is why I recommend the manuscript for next major revision. My detailed remarks are attached in pdf file.

Author Response

Please refer to the attachment for reply content.

Reviewer 3 Report

I read carefully the revised manuscript number:water-1946394, the revised manuscript entitled: "Genesis of carbonated hot springs in Xining Basin and its geothermal significance". Unfortunately, the authors either did not respond to the reviewers' comments at all or responded in part, and the manuscript was not corrected. Check the English Grammar. The English language is moderate. Please check all parts of the manuscript and correct grammatical errors. The authors should ask the help of native English speaking proofreader, because there are some linguistic mistakes that should be fixed. Nevertheless, the manuscript is not acceptable in its current form. Since the quality of the study is lower than average for the publication in the journal, tools for objective function optimization are unclear in the methodology and conclusion, but it still needs a major revisions before reconsideration. I attached my reviewer comments in the PDF file. Authors should apply all of my comments.

Author Response

(The authors gave the same response as above.)
